# INSES: Intelligent Navigation and Similarity Enhanced Search for Knowledge Graph Reasoning

## Abstract

GraphRAG is increasingly adopted for converting unstructured corpora into graph structure, enabling relational, multi-hop reasoning beyond chunk-level retrieval. Most systems then reason over these graphs with classic graph algorithms. However, such traversal, tied to static connectivity and 'connected triple' paths, frequently misses latent semantic links in real-world knowledge graphs (KG) that are noisy, sparse, or incomplete. To address this gap, we introduce INSES (Intelligent Navigation and Similarity Enhanced Search), a dynamic graph-reasoning framework that couples LLM-guided navigation, which prunes noise and steers triple selection with embedding-based similarity expansion to recover hidden links and bridge gaps beyond explicit edges, turning search from a purely structural walk into semantics-aware multi-hop reasoning. Additionally, since GraphRAG style search generally incurs higher complexity than naïve RAG, we complement INSES with a lightweight router that sends simple queries to naïve RAG and escalates complex multi-hop cases to INSES, balancing efficiency and reasoning depth. Across multiple QA benchmarks, INSES consistently outperforms SOTA RAG and GraphRAG baselines. Results highlight complementary strengths of coarse-grained text retrieval for easy cases and fine-grained triple reasoning for harder ones. On the MINE benckmark, INSES remains robust across KGs produced by KGGEN, GraphRAG, and OpenIE, improving accuracy by 5%, 10%, and 27%. This work opens the door to adaptive, router-backed KG reasoning.

## 1 Introduction

Graph search is a fundamental problem in computer science, with applications spanning knowledge-graph reasoning, social network analysis, bioinformatics, etc. Classical algorithms such as Depth First Search (Yih et al., 2015), Breadth First Search (Sun et al., 2018), and Random Walk (Lao & Cohen, 2010; Ristoski & Paulheim, 2016) are typically adapted to knowledge graphs (KG), operating over entity–relation triples, rather than applied verbatim. Although effective in traditional settings, such adaptations meet a mismatch in real-world scenarios whose semantics extend beyond bare connectivity and whose structure is often noisy, error-prone, and incomplete. A key limitation of traditional search lies in its reliance on static structures and traversal strategies. In real world settings, knowledge graphs (KG) (Hogan et al., 2021; Ji et al., 2021; Paulheim, 2017) and social networks (Newman, 2003; Easley et al., 2010), not only contain attributes (e.g., names, weights) but can also incorporate embedding representations. However, on the other hand, these embeddings, together with the reasoning and decision-making capabilities of large language models (LLM) (Team et al., 2023; Dubey et al., 2024; OpenAI, 2024; ZhipuAI, 2024), open new opportunities for more intelligent and adaptive search: attribute-aware navigation and semantic control.

Recent GraphRAG's style pipelines organize corpora into graph representations to support multi-hop reasoning (Saxena et al., 2020; Procko & Ochoa, 2024; Hu et al., 2025); at the same time, LLM-guided/PPR variants refine traversal (Sun et al., 2024; Ma et al., 2025; Jimenez Gutierrez et al., 2024). Yet in most systems, exploration is still governed by explicit edges and fixed neighborhood budgets, which privileges connected-triplet locality and leaves semantically implied links outside the traversed subgraph. As in the KG application, errors, redundancies, and missing links are unavoidable when extracting structured knowledge from natural language. Even with advanced con-

struction methods such as OpenIE (Angeli et al., 2015), GraphRAG (Edge et al., 2024), or the more recent KGGEN (Mo et al., 2025), which employs iterative LLM-based clustering to reduce sparsity, imperfections remain. As a result, critical relationships may be lost or fragmented between similar but distinct entities. To illustrate this, let us examine an example. For an article titled "The Life Cycle of a Butterfly" in MINE benchmark(Mo et al., 2025), Table 1 shows some of the entity nodes generated when building KGs using KGGEN, GraphRAG, and OpenIE. Across all methods, we observe the presence of many similar entities, such as butterflies, adult butterflies, and female butterflies. During the reasoning process, some characteristics of butterflies can be generalized to adult butterflies, but some characteristics cannot and vice versa. So should these entity nodes be merged in the KG? If they are merged, information loss and errors may occur; if they are not merged, some important information may be missed during the reasoning process. This situation occurs because of the complexity and diversity of natural language. These latent semantic connections cannot be fully captured by explicit graph edges. However, it can be exploited through embedding similarity during search. That is, even butterflies and adult butterflies should be treated as distinct entities, they also share implicit connections. Such latent relationships are not easily captured by explicit graph edges, but they can and should be leveraged during search. By representing entities with embeddings and incorporating similarity-based expansion during search, we can dynamically enrich the graph and surface the hidden links needed for reasoning.

Table 1: Entity Nodes Generated by Different Methods

| Method | Entity Nodes Generated by Different Methods |
|--------|---------------------------------------------|
| KGGEN | ["adult", "adulthood", "antennae", "appearance", "appreciation", "balance", "beauty", "biodiversity", "birds", "body", "butterfly", "camouflage", "caterpillar", ..., **"food"**, **"food source"**, "host plants", **"life cycle"**, **"lifespan"**, **"plant populations"**, **"plants"**, ...] |
| GraphRAG | ["egg stage", "birds", **"adult butterflies"**, "nectar", "insects", "pupa", "host plants", "larva stage", "chrysalis", "metamorphosis", **"female butterflies"**, "pollination", "reptiles", **"butterflies"**, "caterpillar", **"butterfly"**, ...] |
| OpenIE | ["They", "egg to larva", "specific host plants", "journey filled", "third stage", **"butterfly 's life cycle"**, **"lifespan ranging from few days to weeks"**, **"Life Cycle"**, "changes", "laid", "twigs", "Next time", "to prepare for stage of its life cycle", ..., **"short lifespan ranging"**, **"lifespan ranging from days"**, **"prepare for stage of its life cycle"**, **"lifespan ranging from days to weeks"**, **"life cycle"**, ...] |

Building upon these considerations, we propose INSES (Intelligent Navigation and Similarity-Enhanced Search), a dynamic graph-reasoning framework that do the better reasoning over the graph. To counter the static-connectivity bias and reduce noise, an LLM navigator selects and prunes adjacent triples at each step, steering exploration toward evidence that answers the query rather than exhaustively walking neighborhoods. To mitigate incompleteness and aliasing, embedding-based similarity expansion temporarily augments the frontier with semantically proximate nodes, recovering hidden links not realized as explicit edges. These two components act in tandem, navigation prunes and guides; similarity recovers and connects, turning traversal from a purely structural walk into semantics-aware multi-hop reasoning over imperfect property graphs. To cap cost and avoid drift, INSES runs for a bounded number of iterations (small-world (Milgram et al., 1967) motivated), and we introduce a lightweight router: straightforward queries are answered with naïve RAG, while complex or low-confidence cases are escalated to INSES, balancing efficiency and depth.

We evaluate INSES on three multihop QA benchmarks and observe consistent gains over strong RAG and GraphRAG baselines across metrics, demonstrating robustness to dataset difficulty and reasoning depth. An ablation study shows that similarity-based expansion is the dominant contributor to accuracy, while a lightweight router provides further lift and helps contain cost. Moreover, routing analysis indicates that many shallow queries are efficiently handled by naïve RAG ($\approx 86\%$ on HotpotQA), reserving INSES for complex cases, aligning accuracy with efficiency. Finally, on the MINE benchmark, INSES remains effective across KGs built by KGGEN, GraphRAG, and OpenIE, improving mean accuracy by 5%, 10%, and 27%, respectively. We summarize the main contributions of this work as follows:

- We diagnose why explicit-edge, static-strategies exploration under-captures cross-entity relations in noisy/incomplete KGs, motivating dynamic, semantics-aware search that couples structure with similarity.

- We introduce INSES, which fuses LLM-guided navigation with similarity-based expansion for on-the-fly augmentation and controlled traversal over property graphs.

- We design a lightweight router that preserves RAG-level efficiency on easy queries and escalates complex/low-confidence cases to INSES, yielding better accuracy-cost trade-offs.

- We report consistent gains on MuSiQue/2Wiki/HotpotQA, robustness on MINE across KGGEN/GraphRAG/OpenIE, and ablations showing similarity expansion as the main contributor with routing providing additional improvements.

## 2 RELATED WORK

### 2.1 KNOWLEDGE GRAPH REASONING

Reasoning on knowledge graphs traditionally adapts search procedures (e.g., depth-/breadth-oriented traversals, random walks) to operate over entity–relation triples rather than using general-graph routines verbatim, which implicitly assumes that explicit edges are sufficient evidence trails (Wang et al., 2013; Yih et al., 2015; Sun et al., 2018; Lao & Cohen, 2010; Ristoski & Paulheim, 2016). Recent graph-centric pipelines construct or reorganize structure and then guide exploration for multi-hop reasoning: some form hierarchical/summary trees to route queries across levels (Sarthi et al., 2024; Zhang et al., 2025); others induce community-structured subgraphs for summary-centric retrieval (Edge et al., 2024); a third line dynamically constructs KGs and designs adaptive traversal policies (Li et al., 2024; Wang et al., 2024); further variants couple traversal with LLM decision-making (e.g., beam-style selection) (Sun et al., 2024; Ma et al., 2025) or employ importance-biased walks for multi-hop retrieval (Gutiérrez et al., 2024). Despite these advances, exploration is still largely governed by explicit connectivity and fixed local budgets, which under-captures cross-entity evidence and overlooks latent semantic relations (e.g., aliasing among similar-but-distinct nodes) that are not realized as direct triples. To move beyond edge-only locality and reduce noise from imperfect structure, INSES integrates LLM-guided navigation (pruning/steering triple selection using attributes and semantics) with embedding-based similarity expansion (temporarily extending the frontier with semantically proximate nodes to recover hidden links), turning structural walks into semantics-aware multi-hop reasoning under bounded iterations.

### 2.2 RETRIEVAL AUGMENTED GENERATION

Retrieval Augmented Generation (RAG) integrates retrieval into generation to ground LLMs in external knowledge, evolving from early retrieval-based QA (Chen et al., 2017; Karpukhin et al., 2020; Guu et al., 2020) to end-to-end coupling of retrieval and generation (Lewis et al., 2020), with recent advances using LLMs as retrievers (Yu et al., 2023; Sun et al., 2023) and finer retrieval granularity such as propositions (Chen et al., 2024). In practice, RAG spans text-based, KG-based: text-based variants retrieve semantically similar passages (Gao et al., 2023b; Zhao et al., 2024; Xiao et al., 2025; Chen et al., 2025) but can miss deeper relational structure and include redundant context; iterative schemes that interleave retrieval and reasoning (Shao et al., 2023; Trivedi et al., 2023; Wei et al., 2022; Gao et al., 2023a) improve recall yet increase latency and risk error accumulation without a reliable guide. Graph-based RAG offers more interpretable, precise structure (Wang et al., 2024; Liang et al., 2025). Early work injected KG knowledge directly into model representations (Peters et al., 2019; Liu et al., 2020), while more recent approaches augment LLMs externally by translating relevant KG subgraphs into prompts (Wen et al., 2024; Dai et al., 2025; Zhang et al., 2024), while these pipelines inherit KG incompleteness. Together, these trade-offs motivate systems that preserve text-RAG efficiency on easy cases while invoking structured, semantics-aware reasoning when needed. We address this tension with a lightweight router that keeps easy, shallow queries on standard RAG and escalates complex/low-confidence ones to INSES; once escalated, INSES's LLM navigation + similarity expansion directly targets static-connectivity blind spots by leveraging attributes and embedding proximity during search.

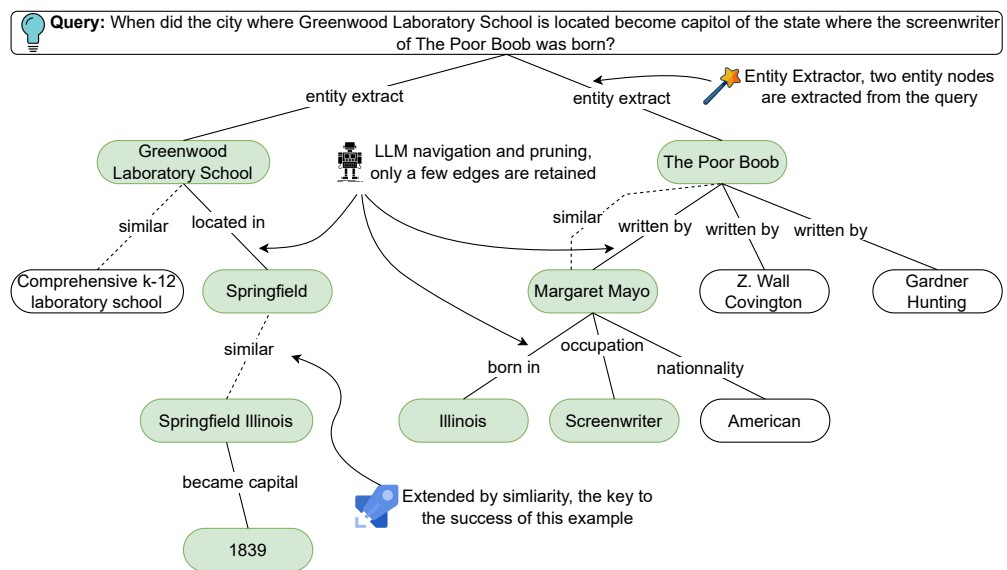

Figure 1: An example of INSES workflow. Solid edges denote explicit relations; dashed edges denote dynamically added similarity edges. Nodes/edges with green background aid answering query. LLM maps query entities ("Greenwood Laboratory School" and "The Poor Boob") to initial nodes, picks relevant triples while pruning noise, and uses similarity expansion to recover latent links (e.g., "Springfield" → "Springfield Illinois"). Navigation and pruning discard spurious edges, while expansion reveals critical connections, together enabling more reliable multi-hop reasoning.

## 3 METHODOLOGY

As discussed above, we introduce intelligent navigation together with similarity-based expansion into traditional graph search to tackle multi-hop reasoning over KGs. We store KG as a property graph(Angles, 2018; Angles et al., 2017). Beyond basic node/edge connectivity, a property graph attaches rich attributes (e.g., textual descriptions, types) to nodes and edges, and each node further has an embedding representation. This allows search to exploit attribute filters and to fuse structural and embedding information, improving both efficiency and accuracy. We begin with formal definitions.

### 3.1 PRELIMINARIE

**Definition 1.** *(Property-Graph-based Knowledge Graph)*

$$KG = (V, E, \lambda_V, \lambda_E, \phi),$$

*where $V$ is the set of entity nodes; $E \subseteq V \times V$ is the set of semantic relation edges; $\lambda_V$ and $\lambda_E$ are attribute functions for nodes and edges; and $\phi : V \to \mathbb{R}^d$ maps each node to a $d$-dimensional embedding. For each edge $e = (u, v) \in E$, the corresponding knowledge triple is $(u, \lambda_E(e), v)$.*

**Definition 2.** *(Multi-hop Search on Knowledge Graphs)*

*Given a natural-language query $q$, multi-hop search (reasoning) aims to identify triples in the graph that are relevant to $q$ and useful for answering it:*

$$\mathcal{T}(q) = \{(u, \lambda_E(e), v) \in \mathcal{G} \mid \text{Relevant}((u, \lambda_E(e), v), q) = \text{True}\},$$

*where $\mathcal{G} = \{(u, \lambda_E(e), v) \mid (u, v) \in E\}$ is the set of all triples, $\mathcal{T}(q)$ denotes the evidence triples for $q$, and $\text{Relevant}(\cdot, q)$ is a relevance function.*

Traditional graph search can be applied to this task, but exhaustive traversal on large KGs is neither computationally feasible nor necessary. Advances in LLM, embeddings, and graph representation

learning enable a more intelligent, dynamic search. Node embeddings let us map nodes to a vector space and expand the graph via similarity; LLMs provide semantic-aware guidance to steer search and prune noise. Building on these ideas, we propose INSES, which couples LLM-guided decision making with similarity-based dynamic augmentation for effective search and reasoning in KGs.

**High-level workflow**. INSES first matches the query to semantically similar entity nodes via vector embeddings. It then iterates: at each step, (i) LLM selects informative triples from the neighbors of the current nodes, triples that directly support answering $q$ or are promising for further exploration; (ii) a similarity module finds nodes most similar to the current nodes. The LLM-selected neighbors and similarity-based nodes are merged to form the next current nodes. These steps repeat until the answer is found or the iteration limits are reached. Figure 1 shows an example workflow of INSES.

### 3.2 STEP 1: EXTRACT INITIAL ENTITY NODES

Use an LLM to extract entities from $q$, that is:

$$\text{LLM}_{\text{Extractor}}(q) = \{m_i\}_{i=1}^k.$$

For each entity $m_i$, retrieve the entity node most similar in $KG$ by cosine similarity to form the initial node set:

$$V_{\text{init}} = \left\{ v_i \mid v_i = \arg\max_{v \in V} \cos\big(\phi(m_i), \phi(v)\big), \ i = 1, \ldots, k \right\}, \tag{1}$$

where $\phi(\cdot)$ denotes the embedding function consistent with the construction of $KG$.

### 3.3 STEP 2: LLM NAVIGATION

In this step, the adjacent triples of the current node, denoted by $T_{\text{adj}}$, are extracted and then pruned by LLM and judged whether they are sufficient to answer the question.

Initialize $V_{\text{current}} = V_{\text{init}}$, $T_{\text{selected}} = \varnothing$. Then

$$T_{\text{adj}} = \{(x, \lambda_E(e), y) \in \mathcal{G} \mid e = (x, y) \in E, \ x \in V_{\text{current}} \text{ or } y \in V_{\text{current}}\}. \tag{2}$$

An LLM acts as a navigator, that is,

$$\text{LLM}_{\text{Navigator}}(q, T_{\text{selected}}, T_{\text{adj}}) \rightarrow \begin{cases} \textsf{STOP}, & \text{if answerable;} \\ (T_{\text{new\_selected}}, V_{\text{candidate}}), & \text{otherwise,} \end{cases} \tag{3}$$

where $T_{\text{new\_selected}} \subseteq T_{\text{adj}}$ are newly selected triples and $V_{\text{candidate}}$ are endpoints of $T_{\text{new\_selected}}$.

### 3.4 STEP 3: SIMILARITY-BASED EXPANSION AND AUGMENTATION

Compute similar nodes for each $u \in V_{\text{current}}$ and keep those above a threshold $\tau_{\text{sim}}$:

$$V_{\text{sim}} = \left\{ v^*(u) = \arg\max_{v \in V \setminus \{u\}} \cos\big(\phi(u), \phi(v)\big) \ \Big| \ \cos\big(\phi(u), \phi(v^*(u))\big) \geq \tau_{\text{sim}} \right\}. \tag{4}$$

Update $V_{current}$ by merging candidates and removing visited nodes:

$$V_{\text{current}} \leftarrow \big(V_{\text{candidate}} \cup V_{\text{sim}}\big) \setminus V_{\text{visited}}.$$

This dynamically augments structure beyond explicit edges to capture latent semantic links.

The complete algorithm is shown in Algorithms 1 in the Appendix B.

### 3.5 COMPLEXITY CONTROL AND ROUTING

LLM-driven navigation introduces additional complexity, so we limit the number of navigation iterations to control cost, by default six, motivated by the theory of small world (Milgram et al., 1967; Kleinberg, 2000). We also introduce a lightweight router that dispatches queries by estimated complexity and confidence: simple queries are handled by a standard RAG pipeline with confidence estimation, whereas multihop queries or cases with low confidence are escalated to INSES for structured graph search and reasoning. This hybrid architecture balances efficiency with reasoning ability. The analysis and demonstration of the routing mechanism and the related Algorithm 2 is shown in Appendix C.

## 4 EXPERIMENTS

### 4.1 DATASETS

To assess the effectiveness of INSES on graph search and reasoning, we conduct experiments on three widely used multi-hop benchmarks: MuSiQue (Trivedi et al., 2022), 2WikiMultiHopQA (Ho et al., 2020), and HotpotQA (Yang et al., 2018). For fairness, we follow the evaluation protocol of previous work such as IRCoT (Trivedi et al., 2023), ensuring that all methods retrieve from the same underlying corpus. To make the experiments computationally feasible while still representative, we sample 1,000 queries from each dataset as our test set.

### 4.2 BASELINES

We compare our approach with three families of baselines. (i) LLM-only methods answer without external retrieval, including Direct Prompting (Direct), the model outputs the final answer without exemplars, and Few-shot CoT Prompting (Few-shot CoT) (Wei et al., 2022), where exemplars provide step-by-step rationales and final answers that the model emulates. (ii) Text-based RAG methods retrieve from unstructured text and condition the LLM on retrieved snippets; we include the standard Naïve RAG pipeline, HyDE (Gao et al., 2023a) (which generates a hypothetical document from the query to guide retrieval), and IRCoT (Trivedi et al., 2023) (which interleaves iterative retrieval with chain-of-thought prompting). (iii) Graph-based RAG methods retrieve and reason over structured representations; we evaluate GraphRAG (Edge et al., 2024), LightRAG (Guo et al., 2024), RAP-TOR(Sarthi et al., 2024) and SiReRAG (Zhang et al., 2025), which leverage graph/cluster structure to aggregate evidence for multi-hop reasoning.

### 4.3 METRICS

We evaluate all methods using two complementary metrics:

**Exact Match (EM).** EM measures whether the predicted answer string exactly matches the ground truth. This is a strict evaluation criterion that rewards only verbatim matches. While widely used in QA benchmarks, EM often underestimates performance when semantically correct answers differ slightly in surface form.

**LLM-as-a-Judge (LLM Judge).** To better capture semantic correctness, we adopt an evaluation protocol in which a LLM acts as a judge. Given the query $q$, the ground truth answer, and the model prediction, the LLM judge determines whether the prediction is semantically consistent with the ground truth and can be considered a correct answer to $q$. This approach mitigates the limitations of surface-level overlap and has recently been shown to be reliable and closely aligned with human evaluation in multiple studies (Gu et al., 2024).

### 4.4 IMPLEMENTATION DETAILS

We follow a standard pipeline for constructing KGs from QA datasets. The constructed KG is stored in the Neo4j graph database (Robinson et al., 2015; Francis et al., 2018). For system integration, we adopt LlamaIndex (Liu, 2022), which offers a modular interface to connect LLMs, databases, and retrieval components in a unified framework. For the embedding model, we use the lightweight model bge-base-en-v1.5 (BAAI, 2024), chosen for its balance between accuracy and efficiency.

Unless otherwise specified, all experiments use GLM-4 (ZhipuAI, 2024) as the LLM backbone for reasoning, navigation, and answer generation. To evaluate the robustness of our approach, we also include ablation studies and comparisons with stronger models - GPT-4o (OpenAI, 2024).

## 4.5 MAIN RESULTS AND ANALYSIS

Table 2 reports the performance of all baselines and our proposed method on three datasets, which can be summarized as follows:

- Our proposed INSES + Router consistently outperforms all baselines on both EM and LLM Judge across all datasets. The strongest baseline, SiReRAG, approaches our scores on Musique but shows a clear gap on 2Wiki and a non-trivial gap on HotpotQA.

- Several graph-based variants (e.g., GraphRAG) fall short of Text RAG in these short, independent QA tasks. One reason is their reliance on cluster/community summaries as the basis for generation, an approach better suited to long, thematically related document sets than to brief factoid questions. In addition, as noted in the Introduction, there is no perfect procedure for text→KG conversion: real KGs are inevitably incomplete and noisy (missing/ambiguous links). Together with the granularity/organization mismatch, these factors imply different applicability regimes rather than an across-the-board advantage for graph methods. This motivates our routing design: since Text RAG is far cheaper than graph pipelines, routing between Text RAG and INSES balances both performance and cost.

- Most Text RAG baselines are relatively stable, and several perform strongly on HotpotQA; for example, Naïve RAG (Top-10) comes close to our method on that dataset. This supports the view that Text RAG excels on simpler or short-chain questions.

- In most cases, LLM Judge and EM track closely. Larger gaps occur primarily on HotpotQA, suggesting that its answer format affects exact string matching more than semantic consistency, making LLM Judge a useful complementary metric there.

Table 2: Performance comparison among baselines and INSES on three benchmark datasets in terms of EM and LLM Judge.

|  | Baseline methods | Musique | | 2Wiki | | HotpotQA | |
|---|---|---|---|---|---|---|---|
|  |  | EM | LLM Judge | EM | LLM Judge | EM | LLM Judge |
| *LLM only* | GLM-4 (Direct) | 0.15 | 0.18 | 0.32 | 0.36 | 0.41 | 0.49 |
|  | GLM-4 (Few-shot CoT) | 0.24 | 0.27 | 0.38 | 0.46 | 0.51 | 0.57 |
| *Text-based* | Naïve RAG (Top-5) | 0.31 | 0.29 | 0.39 | 0.43 | 0.62 | 0.71 |
|  | Naïve RAG (Top-10) | 0.33 | 0.37 | 0.41 | 0.44 | 0.67 | 0.77 |
|  | HyDE | 0.21 | 0.31 | 0.45 | 0.46 | 0.57 | 0.63 |
|  | IRCoT | 0.25 | 0.42 | 0.38 | 0.43 | 0.37 | 0.48 |
| *Graph-based* | GraphRAG (Top-5) | 0.23 | 0.24 | 0.38 | 0.35 | 0.43 | 0.63 |
|  | GraphRAG (Top-10) | 0.26 | 0.36 | 0.50 | 0.43 | 0.47 | 0.61 |
|  | LightRAG | 0.38 | 0.42 | 0.58 | 0.58 | 0.67 | 0.77 |
|  | Raptor | 0.32 | 0.35 | 0.52 | 0.47 | 0.68 | 0.70 |
|  | SiReRAG | 0.44 | 0.43 | 0.48 | 0.53 | 0.61 | 0.75 |
| Ours | **INSES + Router** | **0.46** | **0.47** | **0.67** | **0.71** | **0.68** | **0.80** |

The experiment results highlight the adaptability and robustness of our approach and illustrates the complementary strengths of text RAG and graph-based RAG. Text RAG operates over relatively coarse-grained units (e.g., text chunks) with lower construction and retrieval costs, while KG-based methods operate at the finer granularity of triples, leading to higher construction and retrieval overhead but greater reasoning precision. These results also validate the design of our router mechanism: simple queries can be efficiently handled by Naïve RAG, while more complex multihop reasoning queries benefit from the graph-based search of INSES.

## 4.6 ABLATION STUDY

To better understand the contribution of each component in INSES, we conduct a step-wise ablation study. Specifically, we evaluate the following settings: **(i)** using only the LLM Navigator; **(ii)** adding

Similarity Enhancement on top of the LLM Navigator; and **(iii)** further incorporating the Router. All three variants employ GLM-4 as the underlying LLM. In addition, we test GPT-4o as a stronger backbone to examine the sensitivity of INSES to the choice of LLM.

Table 3 shows that similarity-based expansion makes the largest contribution, yielding substantial improvements of 0.12 (EM) on MuSiQue, 0.07 (EM) on 2Wiki, and 0.05(EM) on HotpotQA. These gains are more pronounced on complex queries, while simpler queries (often ≤2-hop) benefit less since multi-hop reasoning is not required. The router provides additional improvements, though smaller than those brought about by the similarity expansion. Switching from GLM-4 to GPT-4o leads to only modest gains, suggesting that the navigation and similarity enhancement themselves are the dominant factors; once the LLM is sufficiently competent, stronger backbones deliver diminishing returns.

Finally, the HotpotQA results reveal a key insight: naïve RAG already performs well on simpler cases, sometimes outperforming graph search, which highlights the router's particular value. By assigning straightforward queries to Naïve RAG and applying INSES to complex reasoning tasks, the system strikes a balance between cost and performance.

Table 3: Ablation study on three datasets.

| INSES | Musique | | 2Wiki | | HotpotQA | |
|---|---|---|---|---|---|---|
| | EM | LLM Judge | EM | LLM Judge | EM | LLM Judge |
| GLM-4 (Direct) | 0.15 | 0.18 | 0.32 | 0.36 | 0.41 | 0.49 |
| GPT-4o (Direct) | 0.28 | 0.35 | 0.54 | 0.57 | 0.49 | 0.65 |
| Naïve RAG (Top-5) | 0.31 | 0.29 | 0.39 | 0.43 | 0.62 | 0.71 |
| w/ LLM Navigator | 0.32 | 0.35 | 0.57 | 0.51 | 0.53 | 0.62 |
| w/ LLM Navigator + Similarity Enhance | 0.44 | 0.45 | 0.63 | 0.61 | 0.58 | 0.69 |
| w/ LLM Navigator + Similarity Enhance + Router | 0.46 | 0.47 | 0.67 | 0.71 | 0.68 | 0.80 |
| w/ LLM Navigator + Similarity Enhance + Router (GPT-4o) | 0.48 | 0.49 | 0.69 | 0.73 | 0.68 | 0.79 |

## 4.7 ROUTING BEHAVIOR ANALYSIS

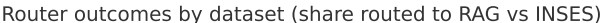
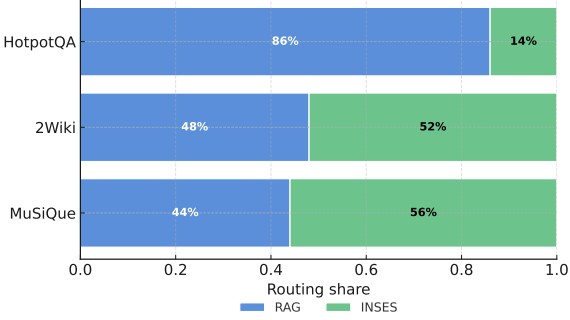

Figure 2: Proportion of queries routed to RAG vs. INSES across three datasets.

To further understand how the router balances efficiency and reasoning accuracy, we analyze the proportion of queries assigned to RAG versus INSES across different datasets. This analysis provides insight into the practical role of the router: whether it effectively delegates simple queries to lightweight retrieval while reserving graph-based reasoning for complex cases.

Figure 2 reports the fraction of queries routed to RAG and INSES on each dataset. The high share of RAG on HotpotQA (86%) indicates that many validation queries can be solved by shallow retrieval; consequently, the marginal benefit of graph search is smaller in this dataset. In contrast, MuSiQue and 2Wiki show near-balanced routing. This observation aligns with our ablation results (Table 3), where similarity-based expansion and multi-hop search yield larger gains on more complex queries.

## 4.8 ROBUSTNESS AND ADAPTABILITY OF INSES

To assess the robustness and adaptability of INSES in KGs of different structure and quality, we evaluate it on the MINE benchmark introduced in KGGEN (Mo et al., 2025). MINE contains 100 articles (each article contains approximately 1,000 words) covering 100 diverse topics, including history, art, science, ethics, and psychology. Each article is associated with 15 factual statements that are grounded in the article.

For each article, three KGs are generated by KGGEN, GraphRAG (Edge et al., 2024), and OpenIE (Angeli et al., 2015), respectively. Then use the native retriever of each method to retrieve supporting triples for the 15 factual statements. An LLM then judges whether the retrieved triples are sufficient to infer the target fact; a query is scored 1 if sufficient (correct), otherwise 0. The accuracy per article is the number of correct queries divided by 15, and we report accuracies across all 100 articles. For comparison, INSES is also run on each of the three KGs, with the same evaluation procedure. To align with KGGEN's setting, we use GPT-4o to judge.

Figure 3 compares the query accuracy distributions of INSES versus KGGEN, GraphRAG, and OpenIE in 100 articles. In terms of mean accuracy, INSES improves by +0.05 on KGs built by KGGEN, +0.10 on GraphRAG, and +0.27 on OpenIE. In particular, KGGEN, GraphRAG, and OpenIE represent three distinct paradigms of KG construction and produce graphs with markedly different structure and quality. However, INSES consistently outperforms each corresponding baseline in all KGs, indicating strong adaptability, due to its effective integration of similarity-based expansion with LLM-guided search navigation.

An analysis of the substantial structural and size differences among KGs constructed by different methods is provided in the Appendix A.

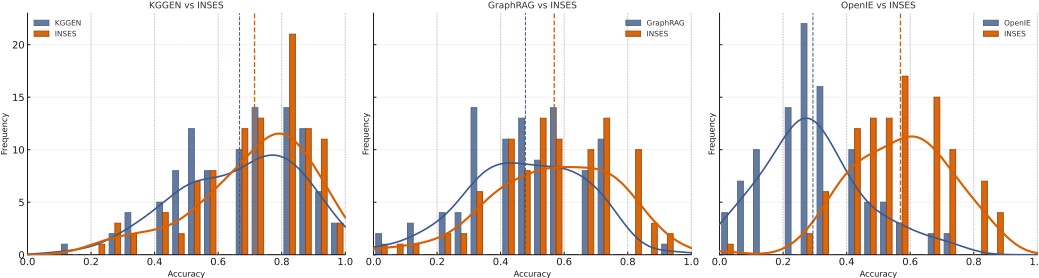

Figure 3: Accuracy distributions comparison across INSES on KGs built by different methods.

## 5 CONCLUSION

In this work, we present INSES, a multi-hop reasoning algorithm for knowledge graphs (KGs) that couples LLM-guided navigation with embedding-based similarity expansion to more effectively search real-world graphs. INSES incrementally selects informative triples while augmenting the local neighborhood with semantically similar nodes, mitigating sparsity and missing links in practical KGs. To balance efficiency with reasoning depth, we introduce a query router that detects easy cases and dispatches them to lightweight RAG, reserving INSES for genuinely multi-hop or ambiguous queries. This modular design keeps efficiency on routine inputs while preserving strong reasoning capability when complex compositional evidence is required.

Extensive experiments demonstrate that INSES consistently outperforms state-of-the-art baselines. Ablation studies confirm that LLM navigation, similarity-based expansion, and the router all contribute meaningfully to performance. These results highlight the adaptability and robustness of INSES for various KG reasoning tasks.

Looking ahead, our work opens up several directions. First, refinement of similarity expansion with stronger noise control and learning-based selection. Second, the similarity expansion and LLM navigation introduced in the INSES algorithm can be used not only in KG reasoning scenarios but also in other graph search scenarios.

## REPRODUCIBILITY STATEMENT

A complete description of the model and retrieval workflow is provided in Section 3 & 4, with the iterative decision loop specified in Algorithm 1 and complexity notes in Appendix B. All hyperparameters, prompts, and inference settings are listed in Section 4 and Appendix, and we report the exact model identifiers and versions for external LLMs/embedding model. Dataset details for MuSiQue, 2Wiki, and HotpotQA, including splits and all preprocessing steps, are documented in Section 4 and Appendix A, and corresponding codes are included in the supplementary materials. Our evaluation protocol (EM, LLM-Judge), decision criteria, and aggregation procedures are described in Section 4. To enable end-to-end replication, we provide an downloadable code archive (supplementary materials) that contains exact configuration files, fixed commit hash, and one-command scripts to reproduce results, as well as an environment specification (requirements.txt) and containers for Neo4j and Qdrant via docker-compose.yml. Instructions for running all experiments with the released processed JSONs and for regenerating results from raw data are included in the README in the supplementary materials.

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

CONTENTS

# APPENDIX

## A  A EXTENDED ANALYSIS OF THE INCOMPLETENESS OF VARIOUS KNOWLEDGE GRAPHS

In the Introduction, we used a concrete example to analyze the differences among knowledge graphs produced by different methods and the issues they entail. We now examine this question at a broader scale and from a quantitative perspective, to better understand the heterogeneity and challenges present in real-world graphs.

Figure 4 compares the average sizes of knowledge graphs generated by three methods on MINE dataset (Mo et al., 2025): KGGEN (nodes=102, edges=72), GraphRAG (nodes=14, edges=13), and OpenIE (nodes=189, edges=265). The spread is substantial: relative to GraphRAG, OpenIE yields about $13.5\times$ more nodes and $20\times$ more edges, with KGGEN in between. The implied average degree ranges from $1.41$ for KGGEN and $1.96$ for GraphRAG to $2.80$ for OpenIE, revealing a clear gradient from sparse to denser graphs. These discrepancies indicate that different extraction paradigms produce markedly different graph topologies: compact graphs risk incompleteness, whereas larger graphs are more susceptible to ambiguity and noise. Consequently, search algorithms for real-world KGs should be both adaptive and robust to such variability, as well as to missing edges and fuzzy surface forms. In this context, combining LLM-guided navigation (semantic filtering, relevance-driven pruning, and error suppression) with similarity-based expansion (to recover latent links, aliases, and paraphrases) is necessary and complementary: the former keeps the search precise, while the latter prevents missed connections in imperfect graphs.

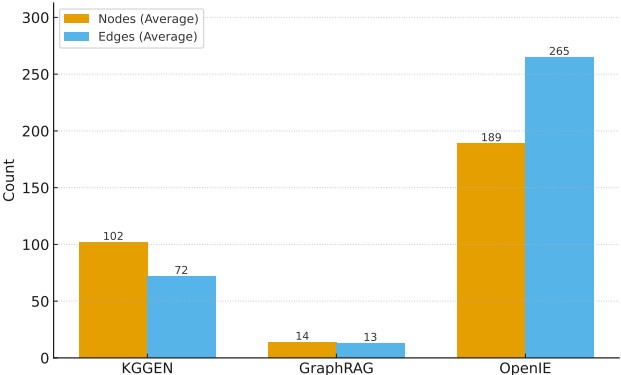

Figure 4: Average size of knowledge graphs generated by three different methods on the MINE dataset (100 articles). Bars report the mean number of nodes and edges per graph produced by KGGEN, GraphRAG, and OpenIE. The large spread across methods highlights the heterogeneity of KGs built from the same corpus, underscoring the need for search–and–reasoning algorithms that are adaptive and robust to incompleteness, ambiguity, and noise.

## B  IMPLEMENTATION DETAILS OF THE INSES ALGORITHM

The INSES algorithm is shown in Algorithm 1.

Equation 1 in Algorithm 1 uses an LLM to extract entities in the query $q$. The relevant prompts are shown in Table 4.

Equation 3 in Algorithm 1 uses an LLM to navigate graph search. The relevant prompts are shown in Table 5.

Table 6 presents a template that allows LLM to answer questions based on retrieved triples.

---

**Algorithm 1** Intelligent Navigation and Similarity Enhanced Search (INSES)

---

Input: Property knowledge graph $KG = (V, E, \lambda_V, \lambda_E, \phi)$, query $q$ stated in natural language
Output: A set of triples $T_{selected}$ that are helpful in answering $q$
  1: Use Equation 1 and query $q$ to establish the initial node set $V_{init}$
  2: $V_{visited} = \{\}$
  3: $T_{selected} = \{\}$
  4: $V_{current} = V_{init}$
  5: **while** $iteration < max\_iter$ and $V_{current} \neq \varnothing$ **do**
  6:     $V_{visited} = V_{visited} \cup V_{current}$
  7:     Get adjacent triples $T_{adj}$ using Equation 2
  8:     Use Equation 3 to let LLM select triples from $T_{adj}$, determine
       whether the current information is sufficient and return $T_{new\_selected}$ and $V_{candidate}$
  9:     $T_{selected} = T_{selected} \cup T_{new\_selected}$
10:     **if** sufficient **then**
11:       break
12:     **end if**
13:     Use Equation 4 to find similar nodes $V_{sim}$
14:     $(V_{current} = V_{candidate} \cup V_{sim}) \setminus V_{visited}$
15:     $iteration = iteration + 1$
16: **end while**
17: return $T_{selected}$

---

Table 4: LLM Extract Entities Prompt

---

**LLM Extract Entities Prompt**

---

Your task is to extract several entities from the given query, so they can be used to search a knowledge graph for clues relevant to answering the query.
Return only the entities you extract, separated by commas, with no other text.

Query: {query}

---

Table 5: LLM Navigation Prompt

**LLM Navigation Prompt**

Your task is to provide support for complex queries and multi-hop reasoning in the knowledge graph. Based on the following query, the visited nodes and the selected triplets, as well as the current nodes and their adjacent triplets, select the triplet numbers (separated by commas) from the adjacent triplets of the current nodes that help answer the query.

Selection criteria: Select the triplets that are most relevant to the query and most likely to help answer it.

Then determine:
1. Based on the visited nodes, the selected triplets, and the triplets you just selected, is this information sufficient to answer the query?
2. If so, answer "sufficient";
3. If not, answer "insufficient";

Your response must be in JSON format with two fields:
"determination": "sufficient/insufficient",
"selection": "triplet numbers, e.g., 1, 2, 3"

Query: {query}

The visited nodes:
{chr(10).join(visited_nodes_info) if visited_nodes_info else 'none'}
The selected triplets and their corresponding source text:
{chr(10).join(all_selected_triplets_info) if all_selected_triplets_info else 'none'}
The current nodes:
{chr(10).join(current_nodes_info) if current_nodes_info else 'none'}
The adjacent triplets and their corresponding source text:
{chr(10).join(current_triplets_info) if current_triplets_info else 'none'}

Table 6: LLM Answer Question With Retrieved Triples Prompt

**LLM Answer Question With Retrieved Triplets Prompt**

You are a helpful assistant that provides accurate and concise answers based on the provided knowledge graph information.
Please answer the following query: {query}

The following information is extracted from a knowledge graph, which contains entities, relationships, and relevant text:
{context}

Your response must be in JSON format with two fields:
1. "reasoning": Your step-by-step reasoning process based on the knowledge graph information. Explain how the entities and relationships help answer the query.
2. "answer": The final answer to the query, as concise as possible without unnecessary explanations.

Example response format:
{{
"reasoning": "Step 1: Identified entity X and its relationship to entity Y. Step 2: Found that entity Z is connected to both X and Y. Step 3: Based on these relationships, concluded that...",
"answer": "Concise answer here"
}}
JSON Response:

## C Text RAG vs. Graph-based RAG, and Why a Router is Sensible

Prior work (Han et al., 2025; Zhou et al., 2025) and our experiments indicate that GraphRAG is not uniformly superior to Text RAG. Table 7 contrasts the two paradigms. The two paradigms differ at a structural level, which naturally leads to distinct strengths and usage regimes.

Table 7: RAG vs. GraphRAG: comparison of pipeline, capabilities, and costs

| Dimension | Text RAG | GraphRAG |
|---|---|---|
| Data form & indexing | Chunk raw text and retrieve via dense vectors; preserves original wording and details | Extract entities/relations or community structure to build a graph, then retrieve by graph and aggregate subgraphs/communities |
| Retrieval characteristics | Strong at in-place factual recall via semantic similarity; sensitive to type words and relation templates; well-suited to short-chain reasoning | Connects evidence across segments via explicit structure; better for long chains/hierarchical reasoning and thematic/context integration |
| Strengths (tasks) | Single-hop and $\leq$2-hop factual QA with rich details; pulling key snippets across documents | Multi-hop ($\geq$3) and long-range reasoning; contextual summarization/thematic synthesis; structured evidence integration |
| Generation trade-offs | Higher context relevance and lower noise; more focused coverage in creative/synthesis tasks | Broader evidence recall and coverage, but more redundancy; typical trade-off: coverage $\uparrow$ vs. relevance $\downarrow$ in creative tasks |
| Efficiency & cost | Low build/query cost; shorter prompts | Higher graph construction cost; retrieval/aggregation tends to inflate prompt length, increasing cost (varies by implementation) |
| Implementation variants | Classic dense retrieval with optional reranking, HyDE, hybrid (sparse+dense) retrieval | KG-style triple retrieval, community-based global/local retrieval, mixed nodes (concepts/passages), etc. |
| Common failure modes | Long-range/cross-document reasoning is hard; chunk boundaries hide global structure | Detail loss/missing or ambiguous links and noise during graph construction can cause retrieval drift; global summarization may lose fine details |
| Routing & hybrid use | Well-suited to handling factual/detail-oriented queries on its own | Well-suited to reasoning/multi-hop queries; can be integrated with or selectively routed alongside Text RAG for complementarity |

**When Text RAG tends to win ($\leq$2 hops).** For most $\leq$2-hop queries, the *answer's immediate neighborhood* is either (i) *explicitly mentioned* in the query, or (ii) *strongly evoked by type/semantic cues* in the query. This explains why Text RAG often suffices:

- **Converging 2-hop** ($A \rightarrow \text{Ans} \leftarrow C$): the answer node is directly adjacent to two entities named in the query. Chunks that mention the answer along with $A$ or $C$ are readily retrieved by dense similarity. For example:

  **Q:** Who authored *Pride and Prejudice* and was the sister of Cassandra Austen?
  **Reasoning:** *Pride and Prejudice* $\rightarrow$ JANE AUSTEN $\leftarrow$ Cassandra Austen.
  **Answer:** Jane Austen.

  Vector retrieval readily surfaces chunks where Jane Austen anchors both query mentions.

- **Chained 2-hop** ($A \rightarrow B \rightarrow \text{Ans}$): even if $B$ and Ans are not named, the query typically carries *type cues* that pull the right evidence in embedding space. Such type/entity clusters

and relation patterns are well captured by modern embeddings, so relevant chunks co-mentioning $B$ and Ans are frequently surfaced. For example:

> **Q:** What river flows through the city that is home to the Eiffel Tower?
> **Reasoning:** Eiffel Tower $\rightarrow$ PARIS $\rightarrow$ SEINE.
> **Answer:** Seine.

The answer is tightly tied to a query cue ("river"), which embeddings capture reliably.

Hence, with appropriate chunking (e.g., 256–1024 tokens with overlap), **Text RAG handles many factual and $\leq$2-hop queries efficiently while preserving fine-grained details**.

**When GraphRAG is needed ($\geq$3 hops / long-range).** As the hop length grows, queries rarely contain all answer-adjacent entities; *explicit multi-hop connectivity* in a KG becomes valuable for exposing long-range correlations. However, the text$\rightarrow$graph step can introduce *detail loss, missing/ambiguous links, and noise*. In practice, search may stall or drift on incomplete graphs, and some facts may be literally absent from the graph, even though they exist in the source text. Therefore, knowledge graphs are more suitable for long-range multi-hop reasoning but at the same time require some enhancement methods, such as property graphs and similarity-based extensions.

**Why a router between Text RAG and GraphRAG.** A *router* lets each method specialize: **Text RAG** serves the abundant simple cases cheaply and with high fidelity to source wording, while **GraphRAG** is reserved for *genuinely multi-hop ($\geq$3) or long-range* problems where explicit structure is advantageous. This not only **balances quality and cost** (simple queries dominate real workloads; GraphRAG is costlier to build/query) but also improves robustness: when type cues suffice, dense retrieval excels; when structural chaining is essential, graph reasoning takes over. In our system, this routing criterion aligns with the structural characteristics above and reflects the empirical boundary between the two regimes.

The Router algorithm is shown in Algorithm 2.

---

**Algorithm 2** Router Algorithm

---

Input: A Naïve RAG system with a vector database, a knowledge graph $KG = (V, E, \lambda_V, \lambda_E, \phi)$, query $q$ stated in natural language
Output: A set of text or a set of triples that are helpful in answering $q$

  1: Use LLM to determine if $q$ is related to multi-hop ($\geq$3) search
  2: **if** False **then**
  3:     route to the Naïve RAG
  4:     Naïve RAG gives an answer with $confidence$
  5:     **if** $confident > Confidence_{threshold}$ **then**
  6:        return the results given by Naïve RAG
  7:     **else**
  8:        route to running INSES on $KG$
  9:     **end if**
10: **else**
11:     route to running INSES on $KG$
12: **end if**
13: return the results given by INSES

---

# D    CASE STUDY OF INSES ALGORITHM

Table 8 is an example of INSES search without similarity expansion.

Table 9 is an example of INSES search with similarity expansion.

Below we analyze a concrete run of the INSES algorithm to illustrate how LLM navigation and similarity expansion work in practice. Table8 shows the execution without similarity expansion, while Table9 shows the full INSES run with similarity expansion enabled. Comparing the two, Table 8 fails to reach the correct answer yet demonstrates the effectiveness of LLM-based navigation;

Table 9 succeeds, highlighting the effectiveness of similarity expansion and showing that during navigation the LLM not only selects relevant information but also filters out errors introduced by similarity expansion. A detailed analysis follows.

From Table 8, the initial entities extracted from the query "Who was the spouse of a leading speaker against slavery and publisher of an antislavery newspaper?" are four: ['leading speaker against slavery', 'antislavery newspaper', 'spouse', 'publisher']. Using embedding similarity, these are matched to four nodes in the graph to form $V_{\text{init}}$: ['Opponent of slavery', 'Anti-slavery newspaper', 'Husband and wife', 'Newspaper publisher'].

In iteration 0, the LLM selects three triples from the neighborhood of these four entity nodes (recorded as $T_{\text{new\_selected}}$). This shows that the LLM navigates and prunes well, avoiding a flood of irrelevant triples. One reason is that rich attribute information in the property graph provides strong support for LLM decision-making; another is that, given the context, choosing relevant clues among available information is not a particularly hard task, so the LLM can keep the search breadth within a reasonable range.

In iteration 1, the LLM selects only one triple (again recorded in $T_{\text{new\_selected}}$): "The North Star $\rightarrow$ Published by $\rightarrow$ Frederick Douglass". Consequently, the candidate set for the next step is a single node, $V_{\text{candidate}} = [\text{'Frederick Douglass'}]$.

In iteration 2, from the neighborhood of node "Frederick Douglass" the LLM again selects only one triple—"The North Star $\rightarrow$ Published by $\rightarrow$ Frederick Douglass"—which had already been visited before. In other words, the opposite-end node of this triple has already been explored. No new candidate nodes are generated in this round, i.e., $V_{\text{candidate}} = \varnothing$. The search therefore terminates without finding the correct answer. Overall, the process shows that LLM navigation is efficient and does not select excessive irrelevant information.

Now consider the process in Table 9. Iterations 0, 1, and 2 proceed similarly to Table 8 but with similarity expansion applied at each round. The LLM's core selections remain essentially the same as in Table 8, and it promptly filters out errors introduced by similarity expansion. In Table 8, iteration 2 produces no new candidates and the search stops. In contrast, in Table 9's iteration 2, the current node "Frederick Douglass" yields a new node via similarity expansion—"Frederick Douglass Memorial and Historical Association"—and this newly surfaced node is precisely what leads to the final correct answer.

In iteration 3, the LLM selects the triple "Helen Pitts Douglass $\rightarrow$ Created $\rightarrow$ Frederick Douglass Memorial and Historical Association," whose opposite-end node is "Helen Pitts Douglass." In iteration 4, the LLM again selects a single triple—"Helen Pitts Douglass $\rightarrow$ Is $\rightarrow$ Second wife of Frederick Douglass"—which directly points to the correct answer. Note that in iterations 3 and 4 the LLM selects very few triples (only one each time) and is not distracted by irrelevant nodes introduced through similarity expansion; instead, it filters them out in a timely manner. This demonstrates that combining LLM navigation with similarity expansion is highly effective: similarity expansion can surface latent links, while LLM navigation can promptly prune potential errors introduced by that expansion.

An additional observation is that the final triple "Helen Pitts Douglass $\rightarrow$ Is $\rightarrow$ Second wife of Frederick Douglass" implies that "Second wife of Frederick Douglass" is modeled as a node in the KG. This also explains why the process in Table 8 failed to find the correct answer: in the constructed KG, "Frederick Douglass" and "Second wife of Frederick Douglass" are two separate nodes. As noted in the Introduction, it is difficult to convert natural-language information into a perfect KG. In this example, the fact "Helen Pitts Douglass is the second wife of Frederick Douglass" can be represented either as "Helen Pitts Douglass $\rightarrow$ Is $\rightarrow$ Second wife of Frederick Douglass" or as "Helen Pitts Douglass $\rightarrow$ Is the second wife of $\rightarrow$ Frederick Douglass," and both representations are reasonable. Such situations are common in KGs. If search and reasoning over a KG rely only on exact structural links, potential connections may be missed. Introducing similarity expansion is therefore an effective way to mitigate ambiguity and incompleteness.

Table 8: Search without similarity expansion

| Iteration | The relevant status of each iteration |
|---|---|
| Query | Who was the spouse of a leading speaker against slavery and publisher of an antislavery newspaper? |
| Entities | ['leading speaker against slavery', 'antislavery newspaper' , 'spouse', 'publisher'] |
| $V_{init}$ | ['Opponent of slavery', 'Anti-slavery newspaper' , 'Husband and wife', 'Newspaper publisher'] |
| iter=0 | $V_{current}$: ['Opponent of slavery', 'Anti-slavery newspaper' , 'Husband and wife', 'Newspaper publisher']. 
 $T_{new\_selected}$: ['Thomas spottswood hinde $\rightarrow$ Occupation $\rightarrow$ Opponent of slavery', 'The north star $\rightarrow$ Is $\rightarrow$ Anti-slavery newspaper' , 'Enos bronson $\rightarrow$ Was $\rightarrow$ Newspaper publisher']. 
 $V_{candidate}$: ['Thomas spottswood hinde', 'The north star' , 'Enos bronson'] |
| iter=1 | $V_{current}$: ['Thomas spottswood hinde', 'The north star' , 'Enos bronson']. 
 $T_{new\_selected}$: [ 'The north star $\rightarrow$ Published by $\rightarrow$ Frederick douglass' ]. 
 $V_{candidate}$: [ 'Frederick douglass' ] |
| iter=2 | $V_{current}$: [ 'Frederick douglass' ]. 
 $T_{new\_selected}$: [ 'The north star $\rightarrow$ Published by $\rightarrow$ Frederick douglass' ]. 
 $V_{candidate}$: [ ] |
| Answer | Not Found. |

Table 9: Search with similarity expansion

| Iteration | The relevant status of each iteration |
|---|---|
| Query | Who was the spouse of a leading speaker against slavery and publisher of an antislavery newspaper? |
| Entities | ['leading speaker against slavery', 'antislavery newspaper', 'spouse', 'publisher'] |
| $V_{init}$ | ['Opponent of slavery', 'Anti-slavery newspaper', 'Husband and wife', 'Newspaper publisher'] |
| iter=0 | $V_{current}$: ['Opponent of slavery', 'Anti-slavery newspaper', 'Husband and wife', 'Newspaper publisher'].
$T_{new\_selected}$: ['Thomas spottswood hinde → Occupation → Opponent of slavery', 'The north star → Is → Anti-slavery newspaper', 'Enos bronson → Was → Newspaper publisher'].
$V_{candidate}$: ['Thomas spottswood hinde', 'The north star', 'Enos bronson'].
$V_{sim}$: ['Pro-slavery southerner', 'Liberty party paper', 'Husbands and wives', 'Newspaper of record'] |
| iter=1 | $V_{current}$: ['Thomas spottswood hinde', 'The north star', 'Enos bronson', 'Pro-slavery southerner', 'Liberty party paper', 'Husbands and wives', 'Newspaper of record'].
$T_{new\_selected}$: ['The north star → Published by → Frederick douglass'].
$V_{candidate}$: ['Frederick douglass'].
$V_{sim}$: ['Newspaper editor', 'The toronto star', 'Opponent of slavery', 'Federalist party', 'Husband and wife', "Country's newspaper of record"] |
| iter=2 | $V_{current}$: [ 'Frederick douglass' , 'Newspaper editor', 'The toronto star', 'Federalist party', "Country's newspaper of record"].
$T_{new\_selected}$: ['The north star → Published by → Frederick douglass'].
$V_{candidate}$: [ ].
$V_{sim}$: [ 'Frederick douglass memorial and historical association' , 'Weekly newspaper', 'Federalists', 'Newspaper of record'] |
| iter=3 | $V_{current}$: [ 'Frederick douglass memorial and historical association' , 'Weekly newspaper', 'Federalists'].
$T_{new\_selected}$: [ 'Helen pitts douglass → Created → Frederick douglass memorial and historical association' ].
$V_{candidate}$: [ 'Helen pitts douglass' ].
$V_{sim}$: ['Frederick douglass', 'English language weekly newspaper', 'Federalist party'] |
| iter=4 | $V_{current}$: [ 'Helen pitts douglass' , 'English language weekly newspaper'].
$T_{new\_selected}$: [ 'Helen pitts douglass → Is → Second wife of frederick douglass' ].
$V_{candidate}$: [ 'Second wife of frederick douglass' ]. |
| Answer | Helen Pitts Douglass |

# E    IMPLEMENTATION DETAILS OF USING LLM AS QUESTION ANSWERER

For clarity about the experimental baselines, we also provide exact prompts. Table 10 lists the LLM only (Direct) prompt, and Table 11 lists the LLM only (Few-shot CoT) prompt.

Table 10: LLM only (Direct) Prompt

---

**LLM only (Direct) Prompt**

---

You are a helpful assistant that answers questions based on your own knowledge.

Question: {question}

Please provide your response in the following JSON format:

"answer": "Your final answer"

---

Table 11: LLM only (Few-shot CoT) Prompt

---

**LLM only (Few-shot CoT) Prompt**

---

You are a helpful assistant that answers questions based on your own knowledge. Below are several examples of chain of thought. You can refer to these examples to think about the question and give the correct answer.
Your answer must be returned in JSON format with two fields: "reasoning" and "answer". The "reasoning" field should contain your step-by-step reasoning process, and the "answer" field should contain the final answer. The "answer" field should be as concise as possible and should not contain unnecessary explanations.

Examples of Chain of Thought:
Q: What language is primarily spoken in the country whose capital is Madrid?
A: First, the country whose capital is Madrid is Spain. Second, the primary language of Spain is Spanish. The answer is {Spanish}.
Q: Who painted The Starry Night and famously cut off part of his ear?
A: First, The Starry Night was painted by Vincent van Gogh. Second, the artist who cut off part of his ear is Vincent van Gogh. The answer is {Vincent van Gogh}.
Q: What continent contains the country whose capital is Nairobi?
A: First, Nairobi is the capital of Kenya. Second, Kenya is located in Africa. The answer is {Africa}.
Q: Which composer wrote The Magic Flute and was born in Salzburg?
A: First, The Magic Flute was composed by Wolfgang Amadeus Mozart. Second, Mozart was born in Salzburg. The answer is {Wolfgang Amadeus Mozart}.
Q: What element has the chemical symbol Fe and is used to make steel?
A: First, the chemical symbol Fe stands for iron. Second, iron is commonly used to make steel. The answer is {Iron}.
Q: Which planet is known as the Red Planet and has the volcano Olympus Mons?
A: First, the Red Planet is Mars. Second, Olympus Mons is a volcano on Mars. The answer is {Mars}.

Question: {question}

Please provide your response in the following JSON format:

"reasoning": "Your step-by-step reasoning process"
"answer": "Your final answer"

---

# F   IMPLEMENTATION DETAILS OF NAÏVE RAG

We implement Naïve RAG using the Qdrant vector database as the storage backend and the embedding model bge-base-en-v1.5. For each dataset, we collect all available context passages, embed them, and store the embeddings in Qdrant. Each context is kept at its original granularity from the dataset; no additional splitting or merging is performed. Table 12 provides the prompt used by RAG at inference time.

Table 12: Naïve RAG Prompt

---

**Naïve RAG Prompt**

---

You are a helpful assistant that provides accurate and concise answers based on the provided context.

Please answer the following query: {query}

Context information is below:
{context}

Your response must be in JSON format with three fields:
1. "reasoning": Your step-by-step reasoning process based on the context.
2. "answer": The final answer to the query, as concise as possible without unnecessary explanations.
3. "confidence": The confidence level of your answer, where 0 means no confidence and 1 means complete certainty. If you cannot derive a reasonable answer from the provided context, the returned confidence level should be low.

Example response format:
{
"reasoning": "Step 1: ... Step 2: ... Step 3: ...",
"answer": "Concise answer here",
"confidence": 0.8
}

JSON Response:

---

# G    IMPLEMENTATION DETAILS OF LLM AS A JUDGE

We employ LLM-as-a-judge in two parts of our experiments. In Sections 4.5 and 4.6, an LLM judge assesses, for each query, whether the answer of a method is consistent with the ground truth; the corresponding prompt is provided in Table 13. In Section 4.8, the LLM judge evaluates whether the triples selected by each method faithfully express the stated fact; the prompt for this setting appears in Table 14.

Table 13: Implementation Details of LLM as a judge in Section 4.5 and 4.6

**LLM as a judge Prompt**

You are an expert evaluator. Your task is to determine if the predicted answer is semantically equivalent to the ground truth answer for the given question.

Question: {question}
Ground Truth Answer: {ground_truth}
Predicted Answer: {prediction}

Instructions: - Compare the predicted answer and the ground truth answer in the context of the question.
- They are considered equivalent if they convey the same meaning, even if the wording is different.
- Respond in JSON format with two keys:
"is_equivalent": true or false,
"explanation": a brief explanation for your decision.

Example response:
{{
"is_equivalent": true,
"explanation": "Both answers correctly state that the capital of France is Paris."
}}

Important: Only output the JSON object and nothing else.

Table 14: Implementation Details of LLM as a judge in Section 4.8

**LLM as a judge Prompt**

You are an evaluator that checks if the Correct Answer can be deduced from the information in the context."

Context:
{context}

Correct Answer:
{correct_answer}

Task: Determine whether the Context contains the information stated in the Correct Answer. Respond with "1" if yes, and "0" if no. Do not provide any explanation, just the number.