# OpenReview forum: "INSES: Intelligent Navigation and Similarity Enhanced Search for Knowledge Graph Reasoning"
_ICLR.cc/2026/Conference — Submitted to ICLR 2026_

### Official Review · Reviewer_NJNK · 2025-11-01

**Soundness:** 3
**Presentation:** 3
**Contribution:** 2
**Rating:** 2
**Confidence:** 4

**Summary:**

This paper introduces Intelligent Navigation and Similarity Enhanced Search (INSES), a dynamic graph reasoning framework designed to improve GraphRAG-style pipelines. INSES combines three ideas:
(1) LLM-guided navigation that prunes irrelevant edges and selects promising triples,
(2) similarity-based expansion that adds edges between semantically similar nodes to mitigate graph incompleteness, and
(3) a router that dispatches simple queries to lightweight text-RAG and complex ones to INSES for efficiency.

This paper evaluate INSES on several multi-hop QA benchmarks (MuSiQue, 2Wiki, HotpotQA) and on the MINE dataset, showing modest but consistent gains over existing text- and graph-based RAG methods.

**Strengths:**

1. The paper identifies a limitation of existing GraphRAG methods that strict reasoning reliance on the explicit graph edges.
2. The combination of LLM-guided traversal, embedding-based expansion, and routing is modular and intuitive.
3. Multiple datasets, detailed prompts, and case studies are provided, improving clarity and reproducibility.

**Weaknesses:**

1. The performance gains are small (1-2 % improvement) without multi-run averages or standard deviations are reported. It is unclear whether the improvements exceed the stochastic variance inherent in LLM outputs, making it difficult to judge statistical significance.
2. There is a lack of computational and complexity analysis. The similarity-based expansion appears to require pairwise similarity checks among candidate nodes, which is $O(N^2)$ in the worst case. The paper provides no runtime/complexity analysis to show whether these extra computations are justified by the previous mentioned small accuracy gains. I think that could also be the reason that the author needs to propose a router based method to reduce the runtime overhead provided by INSES in simple query.
3. LLM-guided graph traversal and similarity-augmented edge expansion have both extensively appeared in prior work (for example, Think-on-Graph, link prediction in KG, etc). The contribution is mainly an engineering integration rather than a new theoretical or algorithmic insight.
4. While the MINE experiment in the last subsection attempts to validate INSES’s ability to handle incomplete KGs, the setup effectively evaluates link prediction rather than reasoning accuracy. Moreover, the main benchmarks (MuSiQue, 2Wiki, HotpotQA) do not reflect the noisy or aliased KG conditions motivating the method (Graph-RAG constructed noisy KG). This weakens the causal link between the stated motivation and the demonstrated improvements.

**Questions:**

See the weakness before.

---

> ### Author Response · Authors · 2025-11-16
>
> Thank you for your thoughtful review and helpful feedback. We clarify the following points.
>
> (1) Our comparisons include representative works from the past two years, including recent 2024--2025 methods. Achieving improvements across multiple aspects under such conditions is a positive indication. The ablation results in Table~3 further demonstrate that the gains are stable and interpretable.
>
> (2) Regarding the complexity concern of similarity expansion: first, due to effective LLM pruning, the number of nodes retained in each search step can be regarded as approximately constant (i.e., $O(1)$). Second, similarity search can be efficiently handled by graph-based vector indices such as Hierarchical Navigable Small World (HNSW/NSW), which retrieve top-k nearest neighbors in $O(\log N)$ time instead of $O(N)$. These algorithms are widely implemented in mainstream vector databases. Therefore, the complexity of similarity expansion is roughly $O(\log N)$ rather than $O(N^2)$.
>
> (3) Our similarity expansion is not the same as existing methods. Think-on-Graph performs graph search using beam search, while link prediction in KG focuses on statically adding edges during KG construction. Although similarity expansion may appear related to link prediction, there are key differences. Static link prediction can introduce many incorrect edges, harming downstream reasoning. In contrast, INSES relies on the tight coupling of LLM navigation and similarity expansion. LLM navigation prunes irrelevant branches and confines the search to a small region, while similarity expansion primarily helps in difficult cases (as illustrated in Figure~1 and case studies in the Appendix D). Errors introduced by expansion are eliminated early by LLM navigation. Routing is introduced to balance cost and benefit, ensuring that INSES applies graph search primarily in challenging reasoning scenarios rather than trivial cases. Since graph search is significantly more expensive than naive RAG retrieval, this balance is crucial. Many impactful RAG/GraphRAG methods---including Think-on-Graph and link prediction approaches---are also engineering integrations. Even if INSES is considered an engineering integration, we believe it deserves a stronger evaluation.
>
> (4) Our experiments on the MINE dataset evaluate INSES's robustness on different incomplete KGs. We are not aware of benchmarks explicitly designed with synthetic noise for KG reasoning. As stated in the Introduction, real-world KGs constructed from natural language inherently contain noise, redundancy, incompleteness, and erroneous links. Thus, there is no need to artificially create noisy KGs; noise naturally exists in practical KGs. If a KG had no noise or incompleteness, similarity expansion would not be necessary. KG incompleteness is an inherent issue—even as many works advance KG construction quality, noise remains unavoidable. INSES is designed to address this challenge from a different perspective: instead of relying on a perfect KG, it leverages the combination of LLM navigation and similarity expansion to identify noise and handle incompleteness.

---

### Official Review · Reviewer_6VR7 · 2025-11-01

**Soundness:** 2
**Presentation:** 4
**Contribution:** 3
**Rating:** 4
**Confidence:** 3

**Summary:**

The paper proposes a method for query answering (QA) on knowledge graphs that uses an LLM to navigate the graph and find an answer. Crucially, the method is thought to be integrated with techniques that automatically construct a knowledge graph from text. For this reason, there might be nodes in the graph with different labels but referring to the same concept, which would make the task of QA more difficult. Instead of trying to detect such nodes and merge them during graph generation, the authors instead propose using embedding similarities during inference in order to "jump" to nodes that would otherwise not be connected. This in turn is claimed to improved retrieval performances for QA.

**Strengths:**

I have appreciated the presentation of the paper, with a clear writing, self-explanatory figures, and a detailed description of both the proposed method and the experimental setting. Although I am not entirely confident, it seems that the many recent relevant works have been discussed thoroughly in Section 2 and in the rest of the paper.

I believe the contributions of the paper are substantial in this area. In particular, I think this paper provides a different perspective where, instead of trying to improve the construction of the graph index, an improved retrieval system is exploited. However, I believe there are some ablation aspects that should be addressed (see weaknesses).

**Weaknesses:**

The paper claims that pruning allows the method to discard triples that are not useful, which in turn helps removing noisy information (L217-218). The pruning is carried out by the LLM Navigator component, whose ablation w.r.t. other components in shown in Table 3. While I have appreciated the current ablation study, I believe it still misses an important baseline where no pruning is performed. My conjecture is that a huge portion of the improvement over the a Naive RAG baseline is due to (1) the presence of the graph index and (2) the similarity enhance part combined with the naive routing mechanism.

Since asking the LLM to navigate the triples quickly becomes very expensive with the average nodes degree, I believe the authors should perform ablation w.r.t. the LLM-based navigator. Furthermore, comparing with Naive RAG instead of a Graph RAG method does not enable me to disentangle the presence of the graph index and all the other proposed components. In other words, I believe a Graph RAG + Similarity Enhance baseline is missing from Table 3 (+ optionally also the Router).

I am willing to raise my overall score if the authors can provide some results supporting the need of the LLM-based navigator.

**Questions:**

- What method has been used to build the graph index for obtaining the results shown in Table 2 for INSES + Router?

See also my points in the Weaknesses part.

---

> ### Author Response · Authors · 2025-11-16
>
> Thank you for your constructive comments. We offer clarifications as follows.
>
> (1) We did not perform ablations removing LLM navigation and pruning because these components are essential for graph-based reasoning. Without LLM-based pruning, the search degenerates into BFS/DFS-style traversal, which is prohibitively expensive for large-scale KG reasoning. As you noted, the performance gains stem from the combination of graph indexing, LLM navigation, similarity enhancement, and routing.
>
> (2) As mentioned above, LLM navigation is indispensable. Removing it for ablation would reduce the method to graph traversal. While exhaustive graph traversal could yield higher accuracy (since it explores the entire graph), the time cost on large KGs is unacceptable.
>
> (3) Table 3 focuses on ablations of INSES's own components. Other GraphRAG methods might also benefit from similarity enhancement (provided their retrieval module supports it). However, integrating similarity-based expansion and routing into graph search is part of our contribution. In Appendix C, we further demonstrate the rationale behind the routing mechanism. The data in Table 3 also showcases the effectiveness of similarity enhancement and routing mechanisms.
>
> (4) For graph index construction, we use the Neo4j graph database with implementation based on the LlamaIndex framework. During KG construction, the corpus is split into sentence-level chunks; an LLM extracts entities and relations from each sentence, and the resulting nodes/edges—along with their embeddings—are stored in the graph database. As a result, the graph database contains both the full structural information and the node embeddings.

---

> > ### Comment · Reviewer_6VR7 · 2025-11-26
> >
> > I thank the authors for their detailed response.
> >
> > >We did not perform ablations removing LLM navigation and pruning because these components are essential for graph-based reasoning. [...]  the search degenerates into BFS/DFS-style traversal, which is prohibitively expensive for large-scale KG reasoning. [...] As mentioned above, LLM navigation is indispensable. Removing it for ablation would reduce the method to graph traversal.
> >
> > **I honestly do not believe that LLMs are indispensable for graph-based reasoning.** Arguably, the "similarity enhancement" you use can already be seen a way to systematically prune the search space by focusing on certain nodes. To avoid the search complexity becomes prohibitively high, you could experiment with a much simpler baseline where you select the next top-k nodes to explore based on the embedding similarity with respect to input query. This for example does _not_ require an LLM.
> >
> > The above approach would actually be similar to how retrieval works in RAPTOR, which would be a lot more efficient than using the LLM navigator in the first place. From your Table 2, RAPTOR achieves 0.52 (EM) and 0.68 (EM) on 2Wiki and HotpotQA, respectively. One the same data sets and from your Table 3, I see that w/LLM Navigator alone achieves 0.57 (EM) and 0.53 (EM). To me these results suggest that using the LLM navigator is maybe not that essential, and therefore it would make sense to perform ablations for it.

---

### Official Review · Reviewer_dL96 · 2025-11-03

**Soundness:** 3
**Presentation:** 2
**Contribution:** 3
**Rating:** 4
**Confidence:** 2

**Summary:**

This paper proposes a novel Graph-RAG system that aims to fix the shortcomings of previous methods, in that they mostly explore the Knowledge Graphs (KGs) via the pre-existing, static edges. These approaches thus under-capture cross-entity relations, as real-world KGs are often incomplete. The proposed method, INSES, bridges this gap by augmenting the neighborhood with potential missing links predicted from the highest entity similarity matches. INSES also employs an LLM to select which links may be most relevant to the given queries at each exploration step, effectively pruning the search space. To resolve the computational efficiency issue, the authors further propose a router algorithm that only invokes the expensive multi-hop INSES search if the given query seems to be sufficiently complex. Overall, the authors empirically demonstrate the superior performance of INSES on a variety of real-world KG datasets, and through ablation studies, show that the embedding similarity enhancement technique indeed introduces the most performance gain.

**Strengths:**

1. The paper's introduction, related work, and experiment sections are well-written, rendering the insights and ideas behind the proposed method easy to understand.

2. The experimental part is extensive and comprehensive, with INSES benchmarking against a variety of baseline methods on many real-world, challenging KG-based QA benchmarks, showing the robustness of the proposed method. The ablation studies are also targeted, giving convincing evidence that the similarity-based missing-link enhancement contributes the most performance gain.

**Weaknesses:**

Most of my concerns regarding weaknesses are related to the Methodology section, which seems to be too high-level and lacking critical technical details.

W1. Throughout the paper, the success of the proposed method hinges on pre-existing embeddings computed for the entities. However, little attention is given (at least before the Experiment section) to discussing how these entity embeddings are computed, and what properties they should satisfy in order for INSES to work reliably. Are these embeddings purely semantical, based solely on the natural language descriptions and types of the entities? Or are they purely structural, computed only from the KG structures? Or are they a hybrid and a mix of both sources of information?

My understanding from the context (which may be very incorrect) is that the entity embeddings used throughout the paper are purely semantical, obtained via textual embedding methods. If that is true, one critical question needs to be asked: why not use more structural-aware KG embeddings? There is a gigantic corpus of literature on KG embedding methods developed for predicting missing links in KGs. These methods can take in both semantic and KG structural information, which intuitively would better suit the purpose of INSES compared to purely semantic-based embeddings.


W2. In section 3.4: Similarity-Based Expansion and Augmentation, the set of entities with the most similar embeddings, $V_{\text{sim}}$ is added to the set of entities to be explored and expanded next. However, in many real-world KGs, the entities with the most similar semantic embeddings to the current entity are very likely to be merely aliases, likely providing little to no extra information that is beneficial to answering the given query. The procedure thus may waste a lot of iterations on visiting or expanding the aliases, as the total number of allowed iterations is only 6. Thus, why not have the LLM navigator make a judgment on the entities in $V_{\text{sim}}$ as well, to make a decision whether these similar entities are also relevant to the queries?

W3. In Section 3.5, the discussion of the Router design, which should be another integral and significant part of the proposed method, is too high-level and lacks critical technical details. How does the Router decide whether a query is simple or complex? How does it assess its own confidence? How reliable is the estimated complexity and confidence? The paper would benefit if more important technical details were discussed in the main text.


Some other concerns regarding the mathematical writing in the Methodology section:

W4. The equation (4) is quite confusing to read. Is it suppose to say something like this?

$V_{\text{sim}}(u) =  \\{ v \in V \setminus \\{ u \\} \mid \cos ( \phi(u), \phi(v) ) \geq \tau_{\text{sim}} \\} ,$

which simply states that $V_{\text{sim}}(u)$ is the set of entities (apart from $u$) whose embedding cosine similarity with $u$ is above a threshold $\tau_{\text{sim}}$.

In addition, a LaTex math formatting typo on Line 264: The $V_{current}$ should be formatted as $V_{\text{current}}$.

**Questions:**

Q1. (Related to W1) Can the authors elaborate on the design of the entity embeddings? What kinds of information should be contained in the entity embeddings for INSES to work properly? Would a hybrid embedding incorporating both semantic and structural information be even more helpful for INSES and boost its performance?

Q2. (Related to W2) Can the authors elaborate on how too many aliases may affect the embedding-similarity-enhancement component? Is it possible for INSES to waste too much iteration budget on expanding aliases?

Q3. In Section 3.1, Definition 1, why define the triplet as $(u, \lambda_E (e), v)$ rather than $(\lambda_V(u), \lambda_E (e), \lambda_V(v))$? More generally, what is the purpose of $\lambda_V$? Since it does not seem that $\lambda_V$ is mentioned or used in any other part of the paper.

Q4. In Section 3.3, is there a restriction on how many candidate endpoints are allowed to be selected by the LLM navigator?

---

> ### Author Response · Authors · 2025-11-16
>
> Thank you for your careful reading and valuable comments. We provide clarifications below.
>
> (1) As you correctly understood, the entity embeddings in our KG are semantic embeddings. As illustrated in Figure 1 (which shows both the search process and a local subgraph), entity nodes such as ''Greenwood Laboratory School'' are encoded by the embedding model and stored in the Neo4j graph database. Structural information, such as the edges in the graph, is also preserved in the graph database. Therefore, the entire KG---including all nodes together with their embedding vectors---is stored in the graph database.
>
> (2) Aliases (or other similar information) indeed play a crucial role in search. For example, in the bottom-left of Figure 1, the nodes ''Springfield'' and ''Springfield Illinois'' are connected by a dashed edge. The role of aliases is similar: they provide potential linking information during search. Although alias-based (or other similarity-based) expansion introduces extra cost and may even lead to incorrect links, such branches are terminated early by LLM-based navigation and pruning, preventing errors from propagating through the search. Therefore, the combination of LLM navigation and similarity-based expansion is essential.
>
> (3) Equation (4) is indeed complex, but your interpretation is exactly what we intended: $V_{\mathrm{sim}}(u)$ is used to find the node that is most similar to $u$ among all nodes (apart from $u$) and whose similarity exceeds a threshold. If the similarity is less than the threshold, it returns an empty value. In Definition 1, we introduce $\lambda_V$ and $\lambda_E$ because our KG is built as a property graph. Unlike a standard graph $G(V, E)$, a property graph attaches additional properties to each node and edge. We use $\lambda_V(u)$ to denote the property information of node $u$, which serves as filtering constraints during search. We appreciate your close examination of these definitions.
>
> (4) In LLM navigation, we typically impose a limit (e.g., 30) on the number of candidate nodes (or edges). In practice, however, the number of nodes selected by the LLM in each step is far below this limit, usually only a few. Given the query and current context, deciding which adjacent nodes/edges are useful is a relatively easy task, and LLMs generally handle it effectively.
>
> (5) Due to space limitations, more analyses and implementation details, including routing analysis and routing algorithms, are provided in Appendix C.

---

### Meta-Review · Area_Chair_JCpK · 2026-01-06

**Summary:**

This paper proposes a dynamic graph-reasoning, combining of LLM-guided navigation and embedding-based similarity expansion. The authors claim that the experiments demonstrate that the proposed method provides empirical gains across several benchmarks and detailed ablation studies show that the effectiveness of the proposed modules.  However, the reviewers’ initial evaluation leaned toward negative and raised several substantive concerns:

1. Lack of analyses and technical details on precomputed entity embeddings and router.
2. Computational overhead caused by similarity-based expansion.
3. The absence of an ablation study, removing LLM Navigation and pruning
4. Marginal performance gains
5. Lack of computational and complexity analysis
6. Limited novelty and contributions

**Reviewer Concerns:**

The authors addressed some of the concerns, but the majority remain partially addressed or unresolved.

1. Analyses and details were provided in Appendix C.
2. The authors acknowledged that the proposed method incurs additional computational overhead but argued that incorrect links can be prevented through pruning or early termination.
3. Reviewer 6VR7 objected to the authors’ rebuttal that LLM navigation and pruning are indispensable for graph reasoning by citing RAPTOR, which employs a more efficient and comparably powerful approach.
4. Partially addressed.
5. A complexity analysis is provided.
6. The authors argued that prior works are not novel and hence their method deserves a stronger evaluation. This limited novelty/contribution is not properly addressed.

**Reviewer Scores:**

The initial ratings leaned toward rejection, and reviewers raised several substantive concerns. The authors addressed some of these concerns in the rebuttal by providing additional analyses and implementation details. However, these additions are insufficient to change the initial evaluation. Especially, the marginal empirical gains, limited novelty, and absence of a key ablation study suggest a recommendation for rejection.

---

### Decision · Program_Chairs · 2026-01-26

Reject